# Colony specificity and starvation-driven changes in activity patterns of the red ant *Myrmica rubra*

**Oscar Vaes**👤*, **Claire Detrain**👤

Unit of Social Ecology, Université Libre de Bruxelles, Brussels, Belgium

* oscar.vaes@ulb.be

## Abstract

Although the activity levels of insect societies are assumed to contribute to their ergonomic efficiency, most studies of the temporal organization of ant colony activity have focused on only a few species. Little is known about the variation in activity patterns across colonies and species, and in different environmental contexts. In this study, the activity patterns of colonies of the red ant *Myrmica rubra* were characterized over 15 consecutive days. The main goals were to evaluate the colony specificity of the activity patterns and the impact of food deprivation on these patterns. We found that the average activity level varied across colonies and remained consistent over 1 week, providing evidence that the activity level is a colony-specific life trait. Furthermore, all colonies applied an energy-saving strategy, decreasing their average levels of activity inside the nest, when starved. Starvation induced no consistent change in the activity level outside of the nest. An analysis of activity time series revealed activity bursts, with nestmates being active (or inactive) together, the amplitudes of which reflected the ants' degree of synchronization. Food deprivation increased the amplitude and number of these activity bursts. Finally, wavelet analyses of daily activity patterns revealed no evidence of any periodicity of activity bouts occurring inside or outside of the nest. This study showed that *M. rubra* ant colonies are characterized by specific activity levels that decrease in response to starvation with the adoption of an energy-saving strategy. In addition, our results help to understand the functional value associated with synchronized and/or periodic fluctuation in activity, which has been debated for years.

## Introduction

The remarkable ecological success of social insects depends on their high efficiency in performing collective tasks, such as the exploration of new areas, the exploitation and sharing of resources, the building of a common nest, and the defense of a common territory. As a correlate, popular beliefs have often acknowledged insect societies as the paradigm of industrious animals being extremely active throughout their life. This has led to some misconception about the actual activity level of insect societies and the distribution of activity among colony members. Several studies have provided striking examples of inactivity, with up to 50–70% of

**Funding:** The author O.V. benefited from the grant n°1.E.106.19F awarded by the "Fonds pour la formation à la recherche dans l'industrie et dans l'agriculture" (FRIA). https://www.frs-fnrs.be/fr/ The funder had no role in study design, data collection andanalysis, decision to publish, or preparation of the manuscript.

**Competing interests:** The Authors have declared that no competing interests exist.

workers reported to be inactive in societies of ants [1–3], bees [4], bumblebees [5], wasps [6], and termites [7]. Furthermore, some worker individuals have been found to be persistently inactive throughout their lives [8, 9], giving rise to several hypotheses (reviewed in [10]) about their functional value namely as a reserve working force [11–13].

A general picture of activity/inactivity patterns in insect societies is difficult to draw, due to the many ways in which activity has been defined and measured in colony-level studies. The quantification of activity/inactivity in ant colonies has been based on the foraging effort [14–16], walking activity [17], and walking speed [18, 19] of ant workers, and on the number of pixels changing across successive images of ant position inside nests [20–22]. The rate of between-worker interaction [19] and the time spent working on tasks [8, 23, 24] have also been used as proxies for the activity level.

Regardless of the operational definition used for the quantification of activity, differences in activity level are expected across colonies of the same species. This variability can reflect fluctuations in the biotic and abiotic environments, with some effects being fairly predictable. The ambient temperature is probably the best-known abiotic factor that determines the activity of ectotherms, with insect societies displaying an optimal range of temperature for the performance of daily tasks and privileged events, such as sexual swarming [25–27]. Several biotic factors, such as the predation risk and presence of competitors, all of which vary throughout the day and across seasons, can generate periods of high and low activity in insect colonies. With the fluctuation of food availability over time, ant colonies generally increase their foraging efforts after prolonged starvation periods [20, 28, 29] (but see [30]) and may alter the daily patterns of activities performed inside the nest, such as the propensity of workers to care for the brood [31]. In starved colonies, foragers can also modulate the laying of the chemical trail [32] and in-nest workers can be more responsive to recruitment signals [33], facilitating the mobilization of workers to new food sources. Finally, the activity levels of hymenopteran colonies can differ intrinsically. Naturally occurring variation in colony foraging activity has been reported in ants [34, 35], wasps [36], and honeybees [37], in which it can also be artificially selected [38].

The periodicity of activity bouts has been examined using activity time series. The best-known periodicity in activity is seasonal [39, 40] and circadian [17, 41, 42]. Fluctuations in light or temperature, related directly to the hour of the day or period of the year, can act as pacemakers that drive the activity of almost all organisms, including insect societies (see e.g., [17, 41]). In addition to seasonal or circadian rhythms, striking activity cycles with a periodicity of only 15–30 min emerge in ant colonies kept under constant and standardized laboratory conditions. Such cycles seem to be generated by the spontaneous self-activation of individuals that transmit their state of arousal through contact with nestmates, rather than by an external pacemaker or environmental change [21]. Activity tends to generate further activity by enhancing encounter rates and movement among workers. The discovery of these short-term periodic activity bouts has raised interesting and still-debated questions about their adaptive value [19, 43–45].

Importantly, such short-term activity cycles have been described in a very limited number of ant species, all belonging to the genera *Temnothorax* and *Leptothorax* [20–22, 46, 47] (but see [17] on *Camponotus rufipes*). The ant genera with ultradian activity patterns studied to date form small colonies with only a few dozen individuals and have slow-paced lives. The extent to which short-term activity periodicity is a trait shared by other ant species remains an open question. Little is known about the generic value of the synchronized activity bouts that emerge spontaneously in ant colonies in the absence of exogenous signals such as the presence of food or light/dark alternation. Whether the average activity level and temporal organization of activity are stable colony-specific traits also remains unknown.

With this study, we aimed to fill this knowledge gap by characterizing the activity patterns of the red ant species *Myrmica rubra*, which deeply differs in its behavioral ecology, its pace of life as well as its colony size, while showing a high diversity of nestmate relatedness across colonies [48–50]. We recorded colony activity over several successive days to clarify the stability and colony specificity of the global activity level. We also investigated the flexibility and resilience of activity levels with exposure to fluctuations in resource availability, nutritional needs, and in-nest tasks to be performed (e.g., food sharing). To this aim we analyzed activity patterns of ant colonies under a stable condition of satiation, a stressful condition of multiple-day food shortage, and a condition of recovered satiation through the exploitation of a newly available food source. All the activity patterns measured inside the nest will be characterized in terms of their average level, periodicity and synchronization of activity bouts. We examined whether a greater need for nutrients in starved colonies led to an increase in foraging activity to find new resources and/or a decrease in activity inside the nest to preserve energy. Specifically, we examined whether a decrease in the average in-nest activity level was compensated by greater synchronization of activity bouts among nestmates to facilitate task achievement [20, 46]. For the recovery of satiation following the discovery of a new food source, we quantified whether an increase in food-sharing tasks was accompanied by an increase in the average in-nest activity level and/or a larger number of activity peaks. The study data provide a general overview of variation in ant colony activity patterns inside and outside of the nest over time, spontaneously and in response to changing food demands and resource availability.

## Method

The studied species is the red ant, *Myrmica rubra* (Linnaeus, 1758) (Hymenoptera: Formicidae). Ant colonies were collected from the year 2018 to 2021, during the summer, in woodlands located at Sambreville (Namur district: N 50˚25.210'; E 004˚37.878') and Aiseau-Presles (Hainaut district: N 50˚25.657'; E 004˚35.764') in Belgium. These field colonies were kept under controlled laboratory conditions (21˚C ± 1˚C; 50% ± 5% humidity and 12L-12D day-night cycle) for a minimum duration of 30 days before carrying out the experiments. Each colony was used no longer than 10 months after its collection in nature. We provided colonies with water, sucrose solution (0.3M) and freshly killed mealworms (*Tenebrio molitor*) *ad libitum*. Thereafter, these field colonies were divided into standardized experimental colonies that consisted in 200 workers, one queen and 20 larvae (from the 1st to the 3rd larval instar). As for the 200 workers, we aimed to standardize the ratio of inner workers and foragers in the experimental colonies by always taking 140 individuals from the nesting tubes of field colonies and 60 individuals from their foraging area.

Each experiment took place in a 22x38cm plastic tray (Fig 1) with border walls covered with Fluon® (polytetrafluoroethylene) to prevent ants from climbing out. The experimental nests were made from 2mm thick, laser-cut Plexiglas squares that provided the ants with a nesting area of 30.25cm² (5.5x5.5cm). The single entrance to the nest was a 1.0cm long and 0.5cm wide opening. The nest ceiling consisted in another Plexiglas square and was covered by a red filter paper to create darkness inside the nest. The nest floor was made of plaster and humidified from below with a cotton wool wick that was connected to a water reservoir. The nests were off centered in the tray and placed 2.0cm away from its back side with the nest entrance facing the remaining space of the tray over which the ants could forage. The nest area was chosen in order to host up to 200 ants without having ants climbing on each other under the nest ceiling while still leaving some space for workers to move inside the nest.

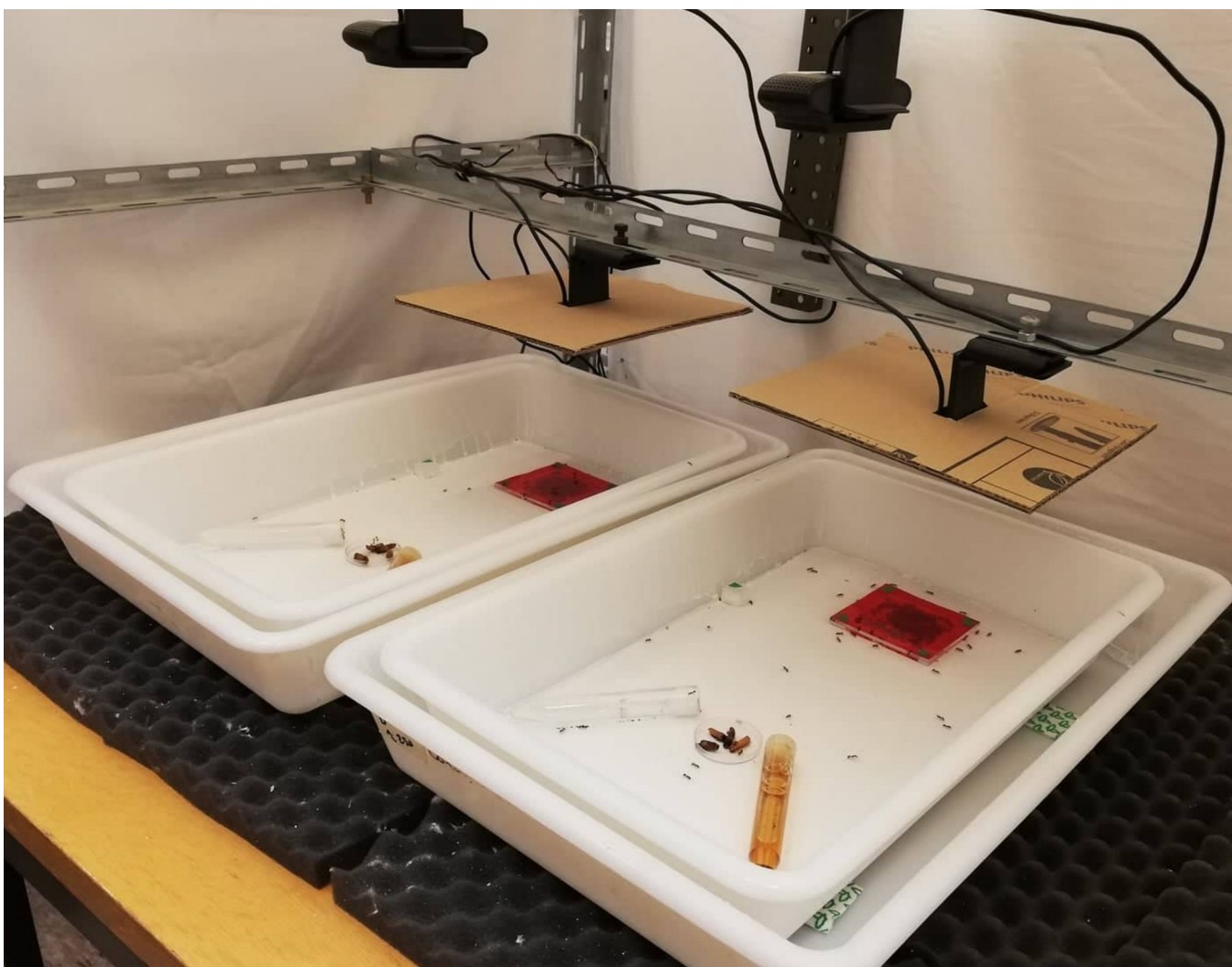

**Fig 1. Picture of the setup with two experimental trays.** Each tray contains the red sheet- covered experimental nest, a sugar tube, a water tube and a Petri dish with mealworms. A camera is set over the nest and another one over the foraging area.

## Experimental procedure

An experiment lasted for 15 days on each tested ant colony. Before the experiment, the ants were placed in the experimental tray, and we let them settle inside the Plexiglas nest for 24 hours before any data collection. For the next 15 days, on each day, we video-recorded the nest interior as well as the foraging area for 2 minutes, every 10 minutes. The experimental trays were illuminated during the whole experiment by a LED lighting in order to get a recording of ant activity that covers a full 24h-period. The experimental colonies were given mealworms, sucrose solution (0.3M) and water ad libitum for the first 4 days (Fig 2). This baseline period allowed us to quantify and analyze the activity patterns of ant colonies under stable conditions of satiation (P1). During these days of colony satiation, new mealworms were provided each day at 10.30am to make sure that the quality of the food stayed satisfactory and similar over time. At 10am on the 5th day of the experiment, we removed the sucrose solution and the mealworms from the foraging area for seven consecutive days. This starvation period (P2) allowed us to detect possible changes in the colony activity due to food deprivation. Both mealworms and sucrose solution were given back to

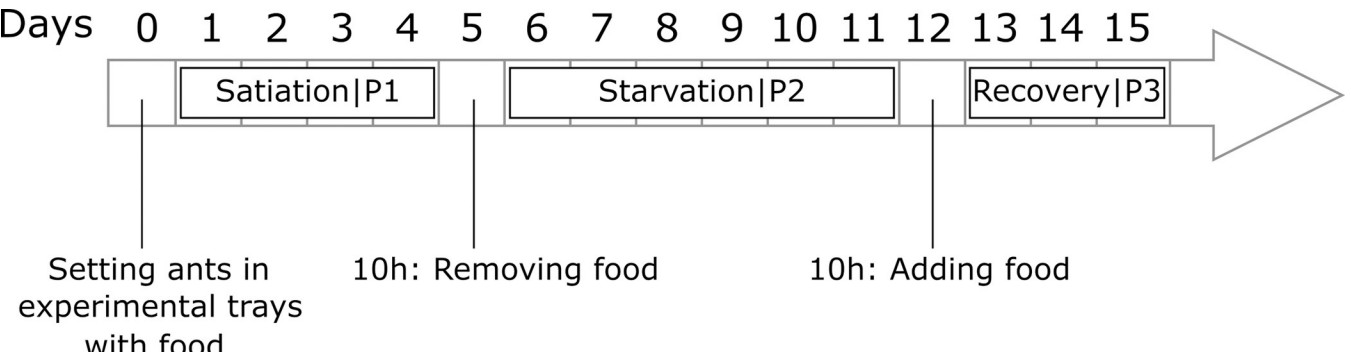

**Fig 2. Timeline of the experiments with three different phases.** Satiation phase = P1, Starvation phase = P2 and Recovery phase = P3.

the colony at 10am on the 12<sup>th</sup> day of the experiment with the ants having access to food ad libitum until the 15<sup>th</sup> day. This recovery period of satiation (P3) lasted three days and allowed to capture the impact of food retrieval and food sharing on the global activity of ant colonies as well to assess the ability of each colony to recover its initial activity baseline at satiation. The experiment was performed on a total of ten ant colonies.

## Data collection

Video recordings were done with the iSpy® v7.1.2.0. software and two Logitech C920® webcams. A first webcam was placed 13cm above the nest ceiling and the second webcam was positioned 33cm above the foraging area (Fig 1). Videos were taken at 15 frames/sec at a 1920x1080 pixels resolution. On each video recording, the level of ants' activity was extracted using OpenCV (Open-Source Computer Vision v4.1.0) library in a Python® (v3.7) script. On video recordings, any movement of the ants resulted in local changes in the brightness of the pixels. The script extracted the number of pixels that changed in brightness between each successive video frame and stored this value as a direct measure of ants' activity. To do so, the image was cropped to keep only the area of interest, here the nest area of 5.5x5.5cm, then was converted into a grey scale to keep only the black ants over a white background. A threshold was set to detect pixels that switched from black to white (and vice versa) and to record it as "activity". The activity measured in the foraging area used the same procedure of automatized image analysis except for the cropping that was focused on a 22x30.5cm area of the experimental tray. Furthermore, we verified that the level of global activity that was automatically detected did not simply reflect changes in the number of ants inside the nest but actual differences in workers' mobility. Thus, during the first phase of satiation, for each colony and each day at 9am, we counted the number of ants inside the nest and the number of foragers. The term "forager" is here referring to any ant observed outside the nest regardless of the task it performed on the foraging area. These numbers were compared to the activity levels measured on the corresponding video recordings made at 9am inside the nest and on the foraging area. Both inside and outside the nest, spearman correlation values were small and not statistically significant (S1 Table), indicating that, when food was given *ad libitum*, changes in the number of ants was not the main explanatory variable for differences in colony activity level.

Ants' mortality was recorded during the whole duration of the experiment. Nine out of ten colonies that were initially tested showed little mortality as well as a similar numbers of dead workers. The average mortality was $0.38 \pm 0.64$ (average dead ant ± standard deviation; n = 9) dead ant per day, which means that on average only $6.1 \pm 2.7$ (n = 9) ants had died per colony over 16 days (24h of habituation and 15 experimental days). For unexplained reasons, one

colony however showed an unusually high level of mortality (38 total dead ants) and had to be taken out of the data analyses.

## Data analyses

All statistical tests were done using the R software v3.3.1. and the significance threshold was set at $\alpha = 0.05$.

Activity inside the nest and in the foraging area were recorded with two-minute-long videos, every 10 minutes. We obtained a total of 144 activity values per day (6 per hour for 24h) and per colony, creating the main database for further analyses of activity patterns.

## Daily activity levels

To compare the average level of activity across colonies and over the course of the experiment, we calculated daily activity indices for each colony. Each daily index was the average of the 144 activity values measured on all two-minute videos that were recorded on this day. Moreover, we estimated to which extent the daily activity levels inside of the nest were linked to those observed in the foraging area by calculating the Spearman correlation value on each colony. This was done only for the starvation period since it was the only phase providing a sufficiently high number of successive daily observations.

We analyzed the activity patterns on each of the three phases of the experiment (Fig 2), the first phase of colony satiation when food was given *ad libitum*, the second phase of starvation when the colonies were food deprived and the third phase of recovery of satiation when food was given back to the colonies. On these daily activity indices, we visually checked on q-q plots the normality of the distribution of residuals. The homogeneity of variance of activity indices was checked by using a Bartlett test. All the distribution of daily indices were homoscedastic for each phase when looking at the activity in the inside and the outside of the nest. We used two-way ANOVAs to test the effect of phase, colony, and interaction of both factors on the daily activity indices. When we found a significant effect, we performed pairwise comparisons by using the Tukey HSD method. Within each phase, we also analyzed how the activity level of one colony changed across the successive days. Therefore, we performed a one-way repeated measures ANOVA to check for significant changes in activity levels over time. When significant, the test was followed by multiple paired t-tests between each day. P-values were then adjusted using the Bonferroni correction for multiple comparisons. These statistical analyses were performed for the activity levels measured inside the nest and also on the foraging area.

## Synchronization of activity

We took a closer look at within-day fluctuations of activity in order to identify moments of synchronized activity/inactivity among nestmates. A high degree of synchronization means that a large proportion of individuals are active (or inactive) at the same time. To estimate the level of ants' synchronization, for each day and each colony, we measured to which extent the activity level obtained at each time point fluctuated around the corresponding average daily value. This was achieved by calculating the coefficient of variation (CV), that is the standard deviation divided by the mean value averaged over the 144 activity values measured on this day. High values of CV corresponded to high numbers of individuals being simultaneously active (or inactive). Previous studies have assessed the level of synchronized motion in other insect species by using a related metric called the index of dispersion (*Temnothorax rugatulus* [47], *Schistocerca gregaria* [51], *Temnothorax allardycei* [52]). Synchronization was analyzed only for ants inside the nest but not for ants located on the foraging area. Indeed, the very low

local density of foragers makes encounters and mutual activation scarcely observed and hence a synchronization of activity unlikely to emerge on the foraging area.

High CV values may result from a few activity bouts of high amplitude but may also reflect frequent small fluctuations that take place all over the course of the experimental day and that are more pertaining to some "activity noise". Thus, we did not limit our analysis of synchronization to the CV metric, but we complemented it by a peak analysis in order to discriminate actual events of workers' synchronization. We used the R function findpeaks to automatically detect the activity peaks and to identify their time of emergence over each daily recording of activity. These functions detected peaks in each 24h-long time series depending on whether the measured activity level exceeded preset threshold values of prominence. Here, a peak was defined as a local maximal value within a set of three consecutive values, where this value had to be at least 10% higher than the average activity index for that day. Prior to the detection of local maxima, the time series was smoothed using a rolling average method with a rolling window of 3. We then counted the number of peaks detected on each day.

All the CV values inside the nest showed a normal distribution and were homoscedastic within each phase of the experiment. Likewise, the total number of peaks for each colony and for each day were normally distributed and homoscedastic within each phase of the experiment. Thus, we performed two-way ANOVAs to test the effect of phase, colony, and the interaction of both factors on the daily CV as well as on the number of activity peaks. When a significant effect was found, we performed pairwise comparisons by using the Tukey HSD method. Within each phase, we also analyzed how the CVs or the number of peaks changed across the successive days by performing a one-way ANOVA for repeated measures. If significant, we carried out multiple paired t-tests between each day whose P-values were adjusted using the Bonferroni correction method.

## Periodicity of activity peaks

A high level of synchronization of ants' activity does not automatically go along with a periodicity of their activity pattern. For example, when a majority of nestmates are simultaneously moving during activity bursts, a colony can be seen as highly synchronized, even though these peaks do not occur at regularly spaced time intervals. In this respect, we carried out wavelet analyses on each daily activity values in order to evaluate the rhythmicity of oscillations of activity inside the nest. This technique, like a Fourier spectral decomposition, provides a breakdown of the dominant periods in the signal that we are studying. However, while a Fourier spectral decomposition transforms the whole signal into oscillatory components, the wavelet analysis fits a short oscillation that is tested at every point of the signal, thus giving it an extra spatial component. Once this oscillation is tested on the entire signal, the process is repeated with an oscillation of longer period and so on until all possible periods are tested. Wavelet analyses are therefore more robust when the signal is non-stationary, i.e., a signal whose properties change over time, for instance, if there is a trend in its mean value. As a result of this wavelet analysis, we obtain the continuous wavelet transform (CWT) that gives the spectral power associated with a certain period, at a certain position of the time series. High power values indicate a stronger contribution of the corresponding period to the oscillations shown by the activity signal. For instance, days where ant colonies showed a high rhythmicity of activity patterns are characterized by high maximum values of wavelet power. After decomposing the periodicity of the signal at each time step, we looked at which period was the most dominant for each colony and each phase. These periods followed a normal distribution and were homoscedastic within each phase of the experiment. Thus, we performed a two-way ANOVA to test the effect of phase and colony to see if the most dominant period was

influenced by the satiation level and if it was colony dependent. As we were interested in ultradian rhythms, we only considered periods that did not exceed 24h. The wavelet analysis was carried out only on the activity patterns observed inside the nest but not in the outside as foragers were too scattered to show any periodic fluctuations of activity.

## Results

### Daily activity levels

We found no significant interaction between colony and experimental phase on daily indices of activity inside the nest (ANOVA: Phase*Colony effect: $F = 1.28$, df = 16, P = 0.225). Interestingly, *M. rubra* colonies significantly differed in their activity level inside the nest throughout the experiment (ANOVA: Colony effect: $F = 37.15$, df = 8, P<0.0001). High levels of activity were consistently found in some colonies such as colony 2 that was always twice more active than colony 7 (Fig 3A). We also found a significant effect of the experimental phase on the activity level inside the nest (ANOVA: Phase effect: $F = 89.82$, df = 2, P<0.0001). During the starvation phase, the daily activity indices were significantly lower (around 20%) than observed in satiated colonies (Tukey HSD: P1-P2 and P2-P3: P<0.0001; full comparisons with confidence intervals in S3 Table). During the recovery phase, just after food was given back to starved colonies, their activity indices increased and even exceeded by around 10% the activity initially during the first phase of colony satiation (Tukey HSD: P1-P3: P = 0.0017; S3 Table). When analyzing separately the first experimental phase, the activity indices of satiated colonies did not significantly change over time (ANOVA repeated measures: $F = 0.52$, df = 3, P = 0.674). Conversely, we observed a significant decrease of activity level over the successive days of starvation (ANOVA repeated measures: $F = 47.15$, df = 6, P<0.0001). Indeed, activity levels were significantly lower after only two days of starvation (S4 Table) with a 36% decrease over the whole period of food deprivation. Once food was given back, ant colonies first displayed the highest level of activity, which then gradually decreased with the recovery of colony satiation (ANOVA repeated measures: $F = 9.08$, df = 2, P = 0.00232) with indices significantly lower on day 15 compared to day 13 (S4 Table).

Outside the nest, the activity levels differed between colonies (ANOVA: Colony effect: $F = 90.86$, df = 8, P<0.0001). The activity of foragers also changed depending on the experimental phases (ANOVA: Phase effect: $F = 27.45$, df = 2, P<0.0001). However, unlike the inner workers, the activity of foragers did not differ between satiated and starved colonies (Tukey HSD: P1-P2: P = 0.372; S3 Table). The main change occurred during the last phase when colonies could again exploit food. Indeed, their activity indices became 28% and 36% higher than in the first and second phase, respectively (Tukey HSD: P1-P3 and P2-P3: P<0.0001; S3 Table). Furthermore, we found a significant interaction effect (ANOVA: Phase*Colony effect: $F = 2.89$, df = 16, P = 0.0007), meaning that certain colonies adjusted their external activity to food availability in a different way. Indeed, when being starved some colonies steeply increased their foraging activity in the outside (e.g., colony 4) and others showed only minimal variations (e.g., colonies 1 and 8). Also, during the last recovery phase, two colonies (5 and 7) did not increase their external activity as observed for the other nests (Fig 3B). After removing these two colonies from the analysis, there was no longer any significant interaction effect (ANOVA: Phase*Colony effect: $F = 1.80$, df = 12, P = 0.062). Looking separately in each experimental phase, we found a significant decrease of the external activity during the first phase of colony satiation (ANOVA repeated measures: $F = 8.9$, df = 3, P = 0.0004) with a lower activity of foragers on day 4 compared to day 1 (S5 Table). During the period of starvation the activity level of foragers slightly changed over time (ANOVA repeated measures: $F = 2.48$, df = 6, P = 0.0357) but no significant difference was found between any pair of starvation days (S5

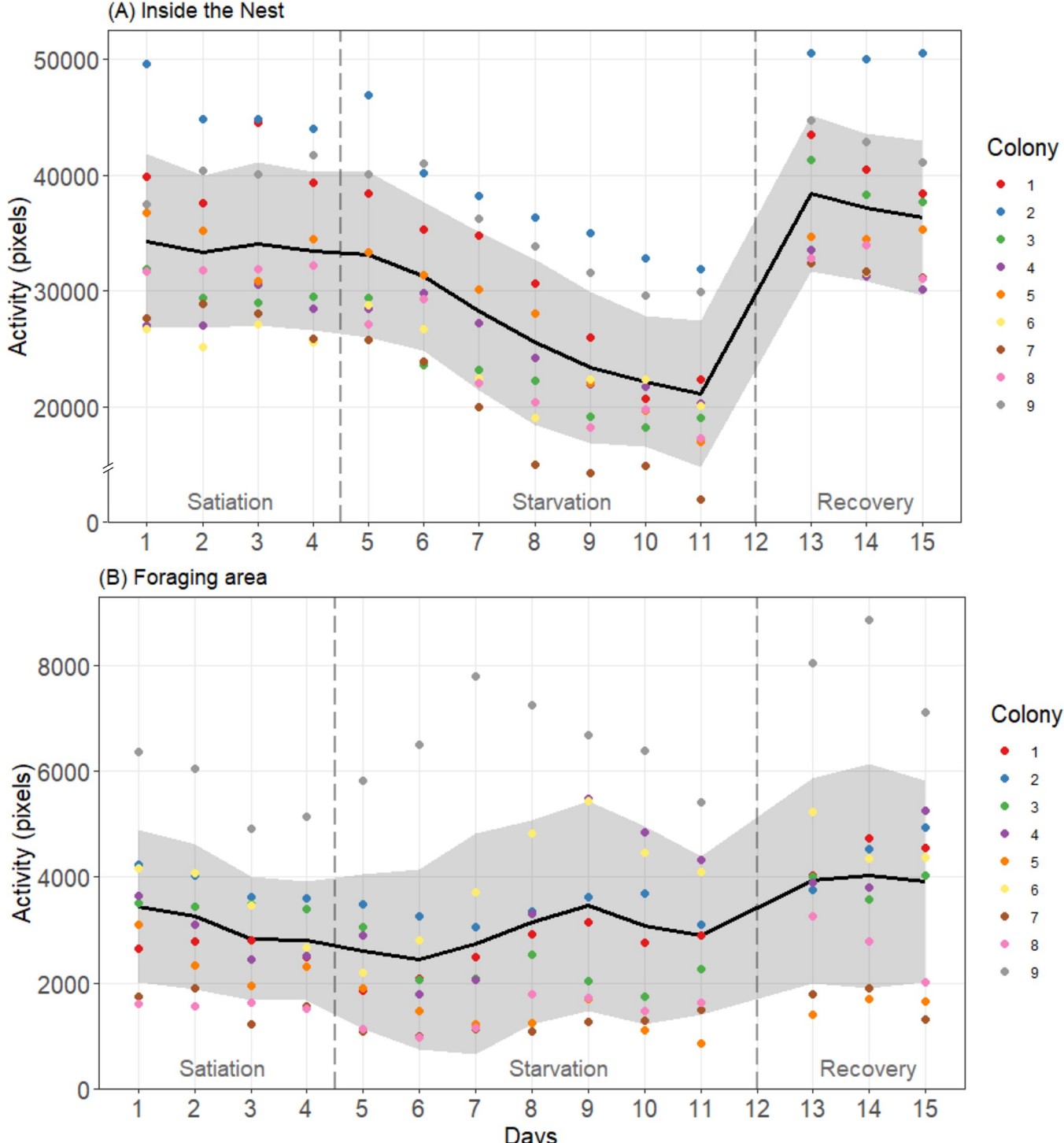

**Fig 3. Time evolution of the daily indices of activity over the experimental days.** Fig 3A shows the activity measured inside the nest and Fig 3B, the activity in the foraging area. The activity indices are expressed in number of moving pixels on the video recordings. Colored points are the daily activity indices measured for each of the nine tested colonies. The black line and the grey area are respectively the average value and the standard deviation calculated on the daily activity indices of the nine colonies. Food was removed on day 5 and was given back on day 12. Vertical dashed lines separate each experimental phase.

Table). Concerning the recovery phase, the daily indices of activity outside the nest did not significantly change over time (ANOVA repeated measures: $F$ = 0.086, df = 2, P = 0.918).

We found no clear-cut relationship between the daily activity indices that were measured inside and outside the nest of the same colony. These two activity values were not significantly correlated excepting for two colonies where a decrease of inner nest activity went along with a higher foraging during the starvation phase (S2 Table).

## Synchronization of activity

For each colony, we examined how the activity values fluctuated across successive measures made every 10min over a 24h-period. Activity bouts of high amplitude reflected a high synchronization of workers that became active simultaneously. On each recording, the time series exhibited considerable variations of activity values from one hour to the next as well as across successive days. The activity patterns also varied between conditions of satiation and food shortage. During the initial phase of satiation, fluctuations of activity values remained of small amplitude and no clear-cut pattern of synchronized activity could be visually noticed (Fig 4A). Even a 24-hour periodicity of activity bouts could not be detected inside or outside the nest. Fig 4 provides an example of activity signals that are typically recorded during the first phase of the experiment, when food was given ad libitum. When colonies were starved, after two or three days of food deprivation, they exhibited quite different patterns of activity over a 24h-period. Beside a decrease of their average activity level, starved colonies showed fluctuations of higher amplitude than those observed in satiated nests (Fig 4B).

When nestmates were highly synchronized, they showed several activity bouts of which the frequent occurrence resulted in high values of CV measured on the corresponding day. Interestingly, we found a significant Colony effect on daily CV suggesting that some nestmates were more likely to synchronize their activity inside the nest (ANOVA: Colony effect: $F$ = 7.5, df = 8, P<0.0001). As illustrated in Fig 5, there was a significant effect of the experimental phase on CVs (ANOVA: Phase effect: $F$ = 59.18, df = 2, P<0.0001) (Tukey HSD: all pairwise comparisons: P<0.0001; S3 Table). Initially, satiated colonies showed moderate bursts of activity with an average CV of 22.6% ± 3.6% (average CV ± standard deviation; n = 36) that did not change over time (ANOVA repeated measures: P1: $F$ = 0.45, df = 3, P = 0.719). When ant colonies were food deprived, these coefficients of variation became higher—even doubled for colony 7—reaching on average a 29.1% ± 8.8% (n = 63) value—(Fig 5). The amplitude of activity bursts, hence the level of synchronization, increased with colony starvation (ANOVA repeated measures: P2: $F$ = 13.94, df = 6, P<0.0001). On the first day of starvation, day 5, the ants in the nest displayed a significantly lower CV than on day 9, 10 and 11 (S6 Table). On day 6, the CV values were significantly lower than on day 9 and 10. The activity became the least synchronized during the last phase. While sharing food inside the nest, nestmates showed only few activity bursts of small amplitude resulting in low CV values of 16.2% ± 3.6% (n = 27) that did not change over time (ANOVA repeated measures: P3: $F$ = 3.23, df = 2, P = 0.0664). We also noticed an interaction effect between the colony and the phase (ANOVA: Phase*Colony effect: $F$ = 2.48, df = 16, P = 0.0033). This interaction effect could be explained by the outlying fluctuations of activity in two colonies (3 and 7) during the starvation phase (Fig 5). Once these colonies were removed from the analysis, there was no longer any interaction effect (ANOVA: Phase*Colony effect: $F$ = 1.01, df = 12, P = 0.445) while the colony and phase effects remained significant (ANOVA: Colony effect: $F$ = 3.47, df = 6, P = 0.0044 | Phase effect: $F$ = 34.17, df = 2, P<0.0001).

## Occurrence of activity bursts

By using a peak analysis, we counted the activity bursts that occurred inside each colony on each experimental day (Fig 4C). Colonies did not differ in their number of daily activity peaks

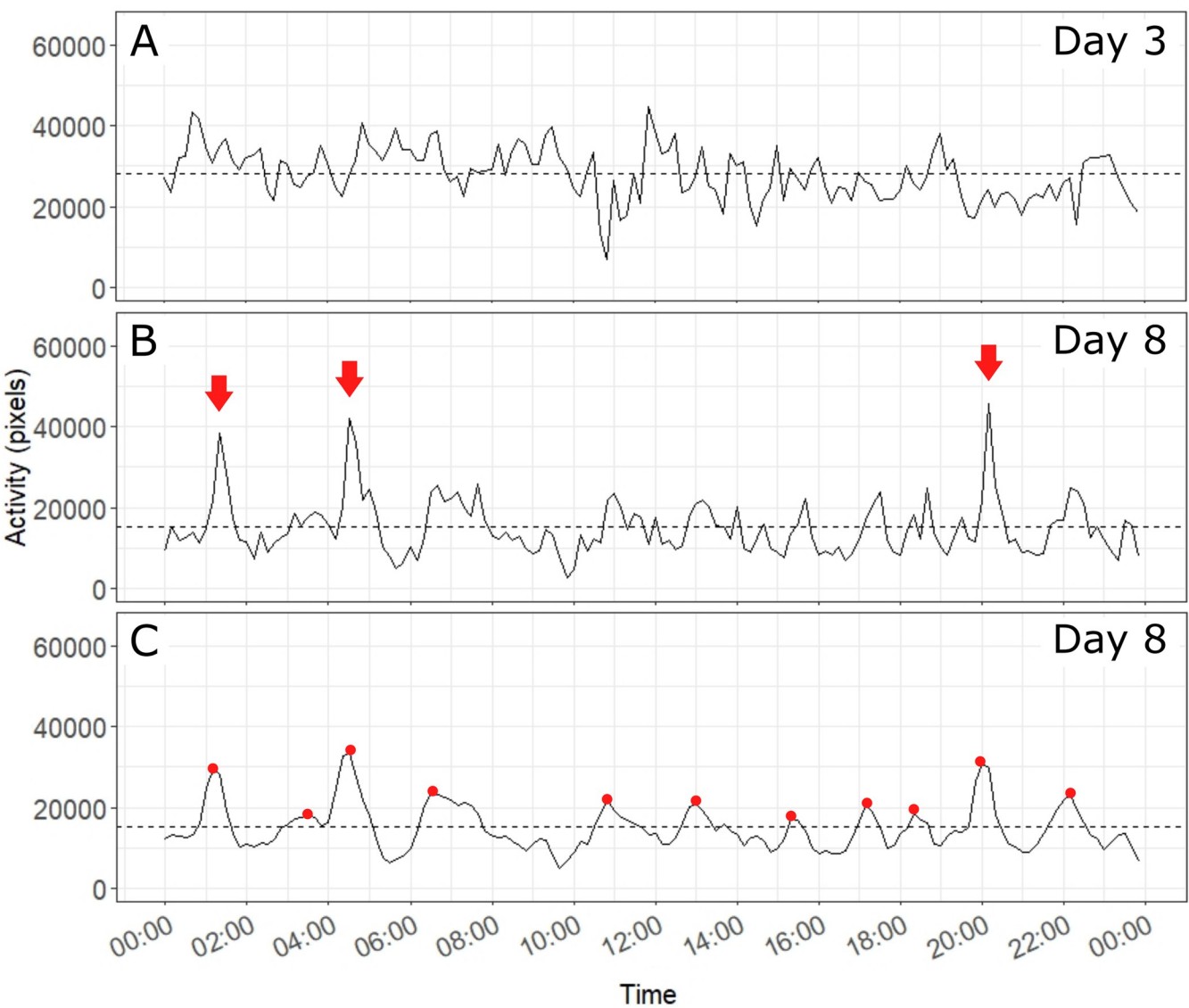

**Fig 4. Examples of activity patterns.** Fig 4A and 4B shows the activity signals of colony 7 recorded respectively on day 3 (satiation phase) and day 8 (starvation phase). Fig 4C represent the activity patterns of colony 7 on day 8 with smoothed signal. The dashed lines correspond to the daily activity index, i.e., the average of all the activity measures made on a day. The red arrows point to clear-cut activity bursts emerging when the colonies were starved. The red dots in Fig 4C show the peaks that were detected by the analysis described in the method section.

(ANOVA: Colony effect: $F = 1.78$, df = 8, P = 0.0901) and there was no interaction effect (ANOVA: Phase*Colony effect: $F = 0.7$, df = 16, P = 0.789). By contrast, the occurrence of activity peaks significantly changed between experimental phases (Fig 6) (ANOVA: Phase effect: $F = 22.2$, df = 2, P<0.0001). During the first phase of colony satiation, colonies showed on average 10 ± 2.3 (average number of daily peaks ± standard deviation; n = 36) activity peaks per day, with no significant change over time (ANOVA repeated measures: P1: $F = 0.175$, df = 3, P = 0.912). By contrast, during phase 2, starvation gradually increased the occurrence of activity bursts (Tukey HSD: P1-P2: P = 0.033; S3 Table). The maximum average daily values of 13.3 ± 1.5 (n = 9) and 11.9 ± 1.6 (n = 9) peaks were reached on days 10 and 11 respectively

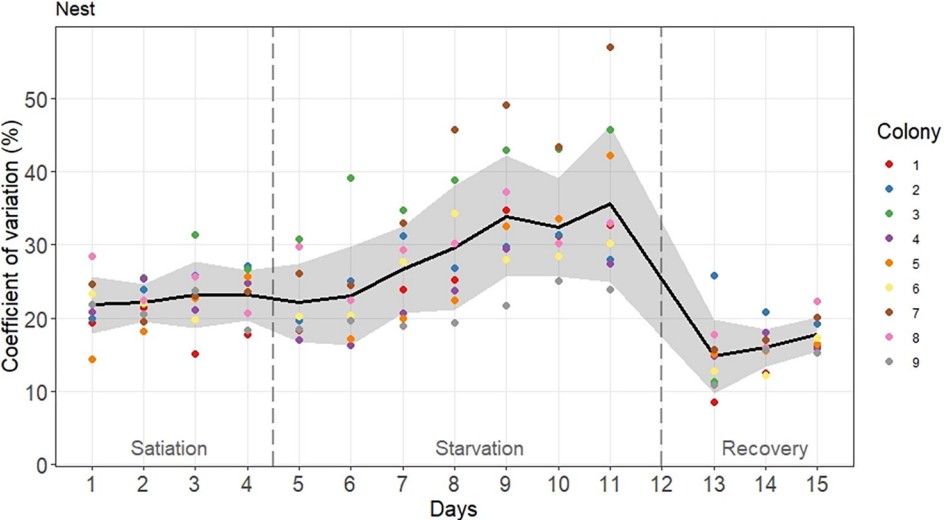

**Fig 5. Time evolution of the CV of activity inside the nest over 15 experimental days.** Colored points are the daily CVs measured for each of the nine tested colonies. The black line and the grey area are respectively the average value and the standard deviation calculated on the daily CVs of the nine colonies. Food was removed on day 5 and was given back on day 12. Vertical dashed lines separate each experimental phase.

(ANOVA repeated measures P2: $F$ = 2.88, df = 6, P = 0.0177) (S7 Table). During the recovery phase, when colonies gained access to food again, the number of detected activity peaks became lower (Tukey HSD: P1-P3: P = 0.0005 | P2-P3: P<0.0001; S3 Table), reached at most 8 daily peaks on average and did not significantly change over time (ANOVA repeated measures P3: $F$ = 3.19, df = 2, P = 0.0684).

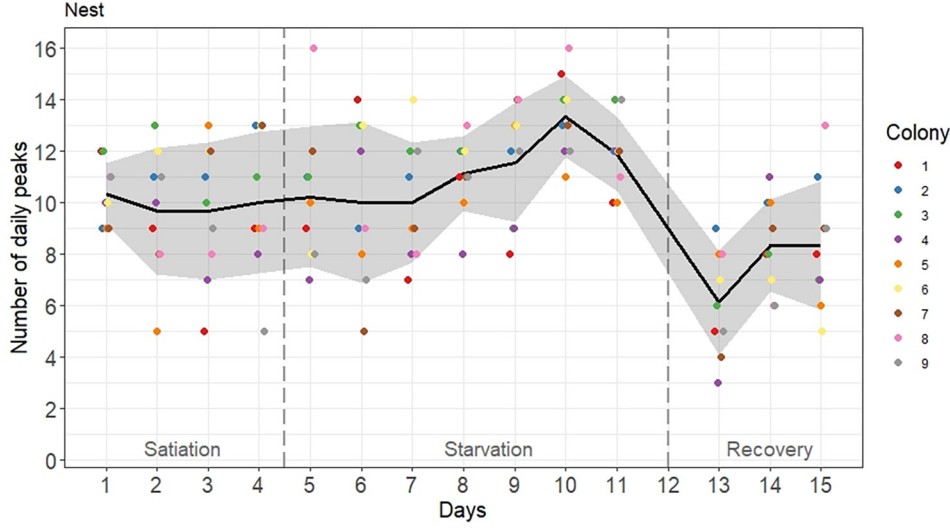

**Fig 6. Time evolution of the number of activity peaks inside the nest over 15 experimental days.** Colored points are the daily peaks measured for each of the nine tested colonies. They are slightly shifted on the x-axis for readability reasons. The black line and the grey area are respectively the average value and the standard deviation calculated on the daily peaks of the nine colonies. Food was removed on day 5 and was given back on day 12. Vertical dashed lines separate each experimental phase.

### Periodicity of activity patterns

Wavelet analyses of time series usually failed to show any activity cycles inside *M. rubra* nests. The wavelet analysis showed many peaks at different frequencies. There is no clear-cut periodicity in the daily patterns of activity. Indeed, out of the nine studied colonies and for the 15 days of experiment, so out of 144 time series, we identified only three occurrences of periodicity that lasted for around 10 hours. These three events of periodicity occurred in colony 1 on day 9, colony 2 on day 6 and colony 3 (Fig 7C) on day 3, which showed a period of around 160min, 180min and 170min, respectively. Moreover, the latter also displayed a periodicity of around 70min, from 11am to 3pm (Fig 7C). Activity cycles were thus scarcely observed and showed no consistency in terms of colony, phase or days involved. Even though there was no-clear cut periodicity of activity patterns, a wavelet analysis performed on each phase allowed to extract the periods that had the overall highest power for this phase. For instance, for colony 7 during the starvation phase, three periods had overall higher power values (Fig 8). Thus, one can also extract, for each phase, the period that was associated to the highest power value (Fig 9). In each phase, the dominant periods markedly varied across colonies. For instance, during the first phase, the dominant period extracted from wavelet analyses ranged from 2h50 (colony 3) to 23h32 (colony 5). Likewise, the dominant period ranged from 1h52 (colony 6) to 22h48 (colony 7) during the second phase and from 5h04 (colony 9) to 23h14 (colony 2) during the last phase. Beside a weak and highly variable periodicity of activity patterns, we found no significant colony effect (ANOVA: Colony effect: $F = 0.83$, df = 8, $P = 0.590$) or effect of starvation on the dominant periods resulting from wavelet analyses (ANOVA: Phase effect: $F = 0.48$, df = 2, $P = 0.625$).

## Discussion

We found that the average activity level inside *M. rubra* nests is a colony-specific trait. Between-colony differences in the activity level persisted over several days: colonies that were most active under the stable satiation condition remained so when being starved and when food was reintroduced. As the experimental colonies contained standardized numbers of queen, workers, and larvae, intercolonial differences in the activity level likely did not stem from differences in brood demand or tasks to be performed inside the nest. Instead, our results suggest that the activity level is specific to each ant colony and consistent across situations–here, along a gradient of food availability and colony starvation. In analogy to the individual-level behavioral syndromes defined by Sih et al. [53, 54], behavioral syndromes, including the carrying over of average activity across different contexts (e.g., in the presence/absence of resources, competitors, or predators), may occur at the colony level. In addition, persistent colony- and regional-level variation in foraging behaviors such as extra-nest activity, exploration, resource discovery and recruitment has been observed in harvester ants [15] and fire ants [35]. Differences in activity among colonies may arise from settlement in natural biotopes with different biotic and abiotic conditions. In the present study, the experimental colonies were maintained in the laboratory for at least 1 month before being tested, thereby reducing the impact of the ants' former living conditions. Colony-level variation in the activity level may also result from the past and current selection of heritable traits related to colony behavior. Variation in activity has been shown to reflect differences across colonies in gene expression in harvester ants [55] and fire ants [14]. Colony activity may also correlate with genetic diversity. For instance, harvester ant queens' engagement in multiple mating promotes genetic diversity in colonies that forage earlier in the day and for longer time periods [56]. In this respect, the ant species, *M. rubra* is characterized by variability in the degree of polygyny and number of patrilines, resulting in pronounced diversity of nestmate relatedness across colonies [48–50].

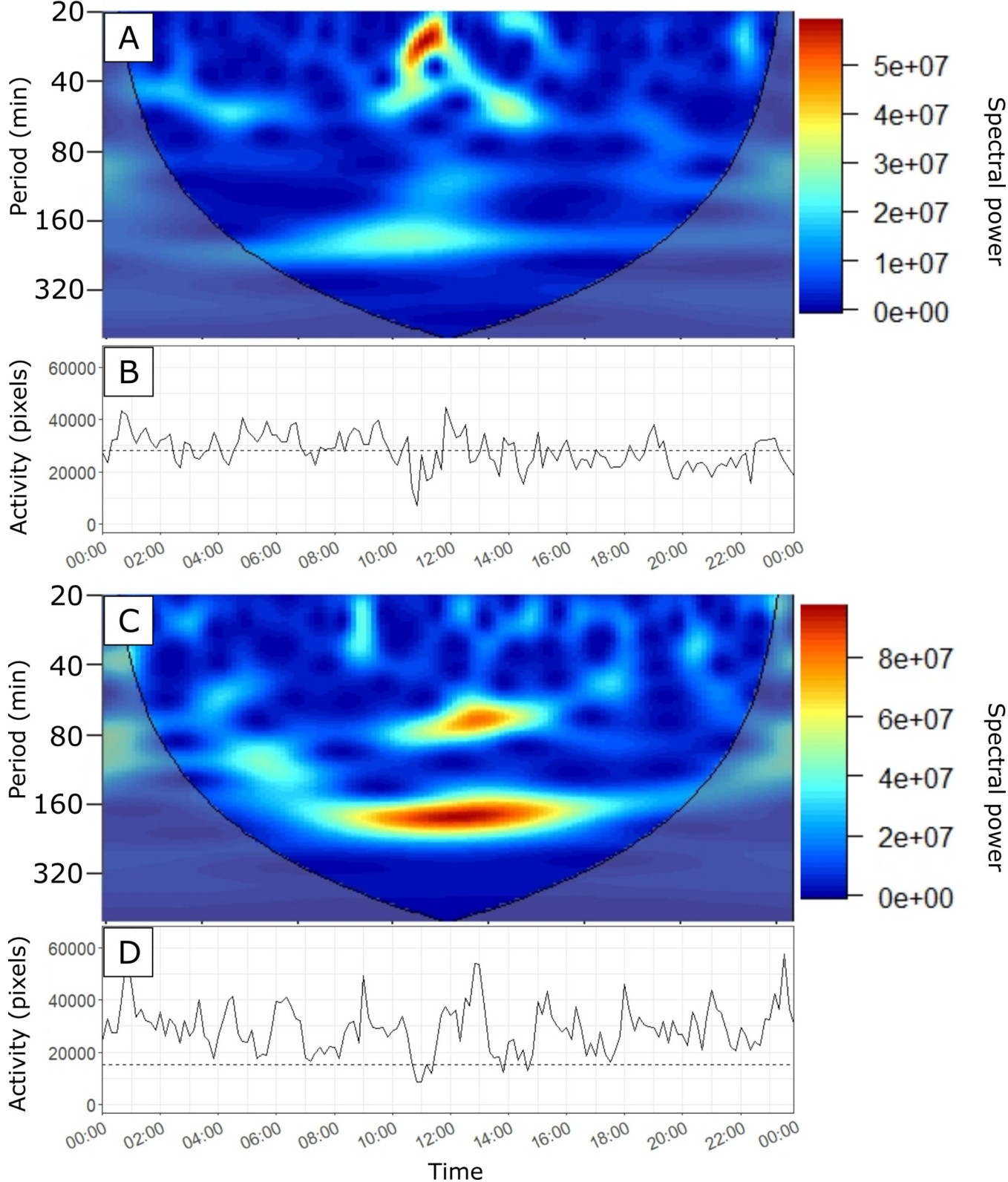

**Fig 7. Graph of the CWT with the corresponding activity signal.** (A and C) are the CWT with the corresponding activity signal below (B and D). The wavelet analysis displays the spectral power associated with a tested period (left y-axis) at a given time position of the signal (x-axis). The two figures on top (A, B) correspond to day 3 from colony 7 and the two figures at the bottom (C, D) corresponds to day 3 from colony 3. (A) The graph of the CWT showed only a brief

timeframe (from 10am to 12pm) during which a period with a high spectral power (red on the right colored scale) was detected. During this narrow timeframe, the activity fluctuated with a periodicity of around 30min. (C) The graph of the CWT showed that the signal was periodic over a longer timeframe of around 8 hours (from 9am to 5pm). In this case, we found a dominant period of around 170min (2h50) that was complemented at 11am by a second shorter period of around 70 minutes.

This diversity may, in turn, at least partially explain differences among colonies in the emergence of collective cooperative behavior and hence in the average colony-wide activity level. Comparative studies of ant colonies founded by lone single-mated queens and those founded by several multi-mated queens would enable the examination of whether a lesser degree of nestmate relatedness is accompanied by a lower average in-nest activity level, in the same way that Wiernasz et al. [56] examined the relationship of workers' relatedness to foraging efficiency.

In this study, starvation was not related to any systematic trend or change in foraging activity outside of the nest. In most colonies, the foraging activity remained unchanged even after a week of food deprivation. Only a subset of *M. rubra* colonies increased their foraging efforts under starvation stress, as reported previously for other ant species [20, 29, 57]. By contrast, all colonies gradually reduced in-nest activity when deprived of food resources. This energy-saving strategy may contribute to colony homeostasis by reducing the global energetic expenses inside the nest to meet lower nutrient incomes due to food shortage. Furthermore, the restriction of activity reduction to the inside of the nest preserves the ability of foragers to engage in

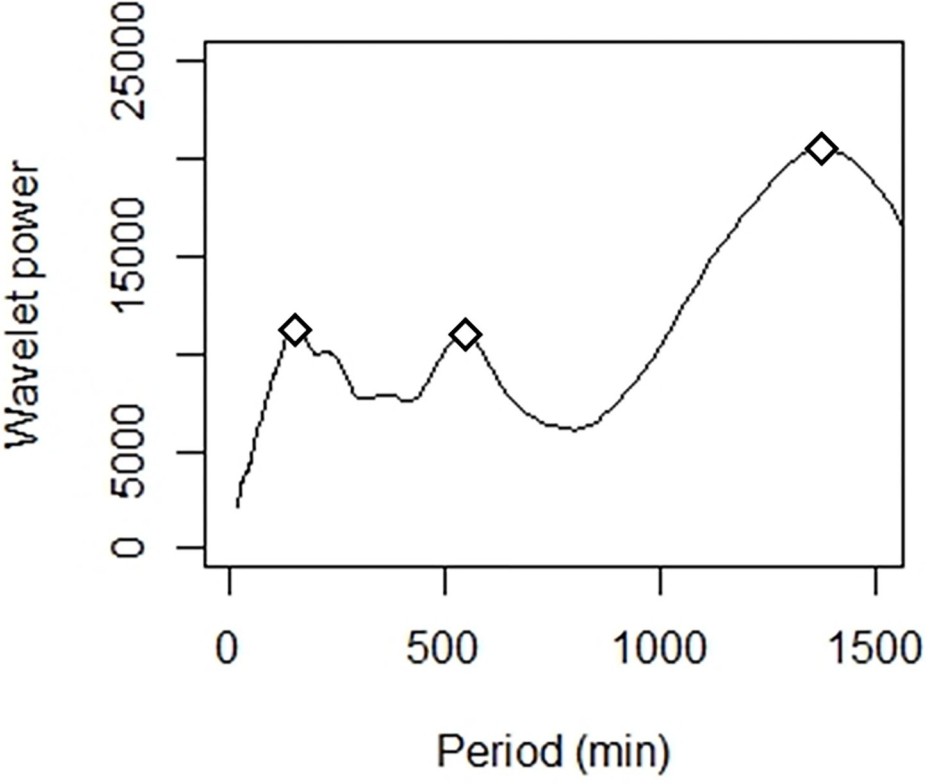

**Fig 8. Distribution of the wavelet power of each tested period for colony 7 during the starvation phase.** The three most prominent periods are highlighted with the black squares. They correspond to periodicities of 2h23 (left), 8h59 (middle) and the most dominant period of 22h48 (right).

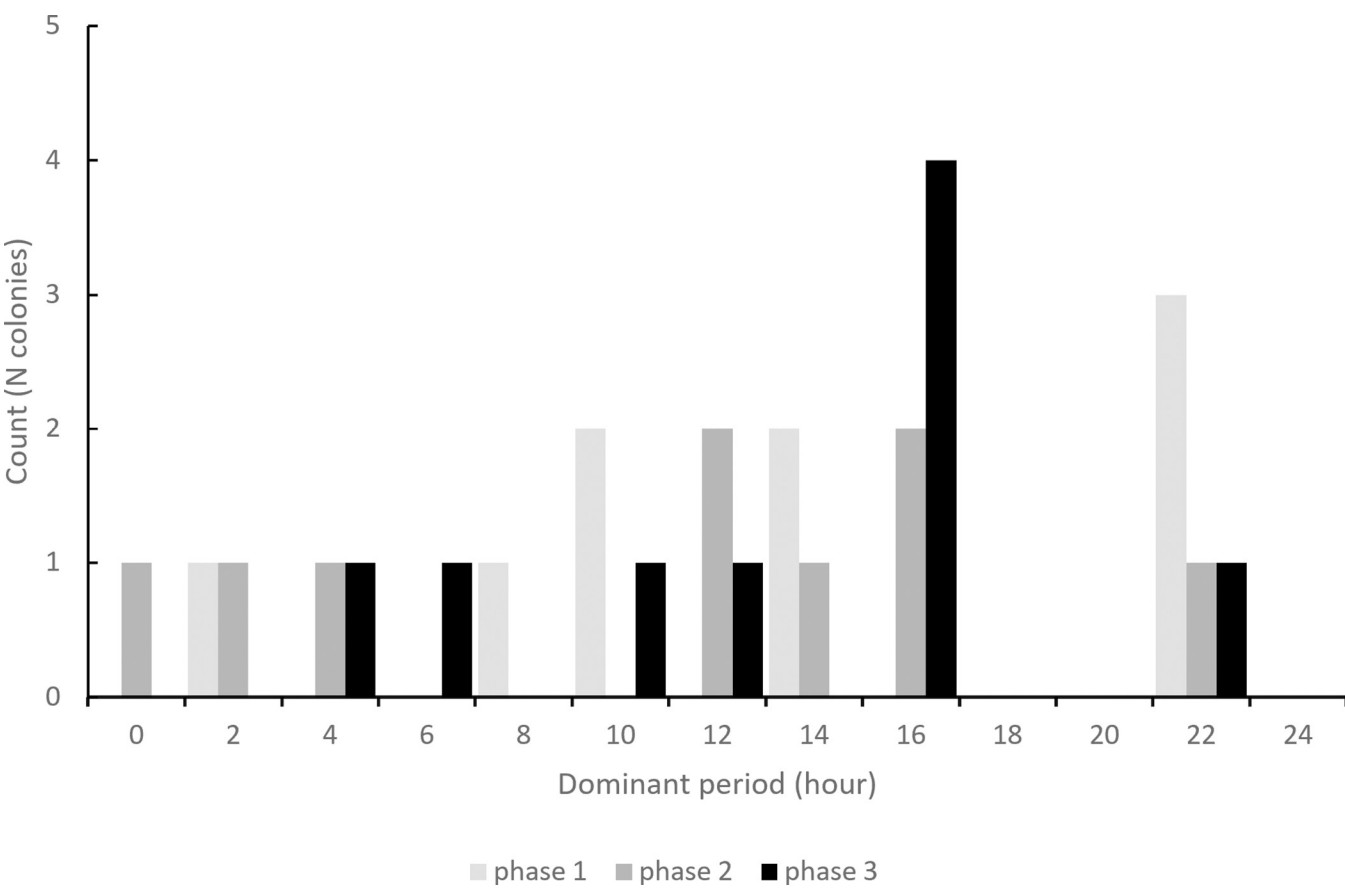

**Fig 9. Distribution of the dominant periods for the activity inside the nest.** Distribution across all 9 colonies and for each experimental phase separately.

external activity to discover new resources. When the starved *M. rubra* colonies regained access to food, the activity levels inside the nest increased steeply, with ant movements likely linked to food sharing, which may involve up to 90% of the worker population as shown in other ant species [58, 59].

A similar activity-reduction strategy has been observed in colonies of the ant species *T. rugatulus* under several months of food deprivation [46], in contrast to its application after only 1 week in the *M. rubra* colonies. This difference could be explained by differences in the species' metabolic rates and hunger levels: the slow pace of life of *Temnothorax* workers probably contributes to their incredible resistance to prolonged starvation, whereas the wakefulness of *M. rubra* workers is accompanied by a higher global metabolic rate, which may have led the colonies to quickly adopt the energy-saving strategy until the return of more favorable conditions. We assume that the lowering of activity as a response to food shortage is a strategy shared by many ant species but operating over very different time scales.

The colony specificity of the activity level demonstrated in this study supports the idea that ant societies act as coordinated, homeostatic systems characterized by specific levels of wakefulness and inactivity. Consistency in the average activity level does not preclude the occurrence of colony-wide fluctuations, including temporary activity bouts (regularly oscillating or not), over the course of the day. Under satiation, the *M. rubra* activity patterns included moments of synchronization, when many ants moved simultaneously. However, these activity peaks did not occur at regular intervals (range, 20 to 640 min). Starvation not only reduced the

average activity level, but also increased the degree of activity synchronization. In colonies starved for 1 week, the number of activity bouts increased slightly while the synchronization metric (i.e., coefficient of variation) nearly doubled. This finding is consistent with the increased degree of synchronization and more frequent activity peaks reported for starved *T. rugatulus* colonies [46], although those colonies showed a greater increase in the number of activity peaks than observed in the present study, likely due to the longer starvation period. Differences in the peak detection technique may also have contributed to this dissimilarity. In the present study, the reintroduction of food to the starved colonies increased the average in-nest activity level and dramatically reduced the degree of nestmate synchronization, resulting in fewer activity bouts of lower amplitude. One possible explanation for these findings is that the signal noise caused by the erratic movements of workers actively sharing food inside the nest masked the detection of activity bursts.

The synchronization of activity found inside *M. rubra* nests is a common trait shared with other ants studied so far (*Temnothorax allardycei* [21], *Leptothorax acervorum* [44], and *T. rugatulus* [47]). The entrainment of nestmates that are roused (become motionless) following contact with active (inactive) workers creates positive feedback loops leading to the emergence of activity (inactivity) peaks. The role of synchronization as an epiphenomenon of mutual entrainment, however, does not prevent this characteristic from having functional value. Some authors point towards activity rhythms serve to regularly lower respiratory carbon-dioxide levels in nests [43]. Others evoke an impact on the colonies' ergonomic efficiency. By being active together may generate local overcrowding and queuing phenomena that reduce task performance efficiency [21] or, alternatively, prompt individuals to search for unperformed tasks, thereby favoring spatially more-homogenous task distribution across the colony [44, 45]. A last hypothesis is that the synchronization of activity may facilitate the transmission of information or materials among nestmates [20]. However, not only information transmission, which is enhanced by synchronized activity peaks, but also information loss, which occurs when an ant forgets relevant information or when chemical information decays, should be considered. The interplay between these two mutually antagonistic feedback loops may lead to the reduction (to zero) or growth (to a steady state) of the population of informed ants. A data-driven model developed by Richardson et al. [19] predicts that information flows through a colony less efficiently when bouts of synchronized activity are periodic. This typically occurs when the lifetime of the information is shorter than the duration of quiescent activity periods, in which case information loss during quiescent periods dominates information spread during active periods. As we found that in-nest activity did not oscillate periodically in the majority of *M. rubra* colonies during starvation or satiation, the flow of information and materials among *M. rubra* nestmates is not expected to be impeded by regularly oscillating activity cycles. At the same time, synchronization of the activity of a sufficient number of *M. rubra* nestmates would enable the efficient completion of cooperative tasks and ensure reliable information exchange (as suggested by Franks and Bryant [60]). Under starvation stress demanding energy preservation, the greater synchronization of workers allows for long periods of inactivity while maximizing task performance and keeping communication efficient during temporary–but not periodic–bouts of activity.

The short (15–30 min) activity cycles of *Temnothorax* colonies serve as an example of self-organized periodicity that is not driven by an exogeneous pacemaker or the inherent rhythmicity of individual ants [21, 61]. This remarkable self-organized behavior may have led to overestimation of the occurrence and importance of such activity cycles in insect societies. Few rhythms were observed inside *M. rubra* nest, and those that occurred were of longer duration (150–180 min) than reported previously. The very low occurrence rate for periodic ultradian patterns suggests that the rhythmicity of activity has little, if any, impact on *M. rubra*

fitness. Similarly, several *Temnothorax* colonies [20, 47] did not display periodic patterns, or displayed more erratic oscillation than others. Furthermore, the periodicity of activity cycles can be disrupted by several social factors, such as the return of foragers [22], location of inter- acting nestmates [22], high density of workers and brood [47], brood developmental stage [62, 63], and ratio of castes within a colony [41]. Regularly spaced oscillations of activity resulting from coupled interactions between workers thus seem to emerge only under very specific con- ditions [21, 64]. Among the many conditions required for the colony-wide propagation of an activation wave at regular time intervals are the appropriate density of mutually entraining ants [21], the homogenous spatial distribution of nestmates with no clear-cut aggregates [22], and a simple nest structure with no compartmentalization. As the natural nests of most ant species lack these traits, activity cycles appear to be striking periodic phenomena whose emer- gence is limited to artificial laboratory conditions. Thus, the rhythmicity of activity appears to be an exception, rather than the rule, in insect societies, and its adaptive value is likely marginal.

Our knowledge about the causes and functional value of activity patterns in ant colonies remains limited, considering the number of unresolved questions and the potential impor- tance of these patterns for the behavioral ecology of social insects. Caution should be taken when attributing adaptive value to activity patterns observed under standardized laboratory conditions. In addition, the functional outcomes of activity levels are difficult to predict, and their impact on colony fitness may depend on the environmental conditions. On the one hand, colony-level variation in extra-nest activity has been found to correlate positively with markers of colony productivity, such as food storage and brood rearing, in honeybees [37] and with population growth in fire ants [35]. On the other hand, an elegant field study of *Pogono- myrmex barbatus* harvester ants showed that an increase in colony activity is not necessarily associated with improved fitness, as less-active and supposedly less-competitive colonies had the greatest reproductive success under harsh environmental conditions [15]. In addition, a better understanding of the effects of the activity level on colony-wide communication and recruitment dynamics, which shape the ability of ant colonies to adjust their responses to changing environmental conditions, is needed [34]. Finally, future studies will have to evaluate whether a synchronization of activity among nestmates actually facilitates the regulation of in- nest tasks, thereby allowing a colony to develop an energy-preservation strategy, namely under conditions of food shortages.

## Supporting information

**S1 Table. Spearman correlation tests between the activity measured at 9am in one zone (nest or foraging area) and the number of ants in the corresponding zone at 9am.** For each colony, tests are performed using all the daily activity indices measured during the satiation phase. P-values in bold are <0.05.
(DOCX)

**S2 Table. Spearman correlation tests between the daily activity indices measured inside and outside of the nest for each colony during the satiation phase (n = 4 per colony) and the starvation phase (n = 6 per colony).** Spearman tests were not performed for the recovery phase due to the too few daily activity indices available per colony. P-values in bold are <0.05.
(DOCX)

**S3 Table. Results of the Tukey HSD test performed on the activity inside the nest, the activity in the foraging area, the coefficients of variation and the number of peaks mea- sured in every pair of experimental phases.** diff = difference in average value between the

two phases. Phase 1 = Satiation phase, Phase 2 = Starvation phase and Phase 3 = Recovery phase. CI = Confidence Interval. P-values in bold are <0.05.
(DOCX)

**S4 Table. Daily comparisons of the activity inside the nest for the starvation phase and the recovery phase separately.** Pairwise t-test comparisons with corrected p-values (Bonferroni method). P-values in bold are <0.05.
(DOCX)

**S5 Table. Daily comparisons of the activity in the foraging area for the satiation phase and the starvation phase separately.** Pairwise t-test comparisons with corrected p-values (Bonferroni method). P-values in bold are <0.05.
(DOCX)

**S6 Table. Daily comparisons of the coefficients of variation for the starvation phase.** Pairwise t-test comparisons with corrected p-values (Bonferroni method). P-values in bold are <0.05.
(DOCX)

**S7 Table. Daily comparisons of the number of peaks nest for the starvation phase.** Pairwise t-test comparisons with corrected p-values (Bonferroni method). P-values in bold are <0.05.
(DOCX)

## Acknowledgments

We would like to thank Bertrand Collignon for his help in developing the python script for the analysis of activity and to thank Luc Dekelver for helping us to collect ants on the field.

## Author Contributions

**Conceptualization:** Oscar Vaes, Claire Detrain.

**Data curation:** Oscar Vaes.

**Formal analysis:** Oscar Vaes.

**Funding acquisition:** Oscar Vaes, Claire Detrain.

**Investigation:** Oscar Vaes.

**Methodology:** Oscar Vaes, Claire Detrain.

**Project administration:** Oscar Vaes.

**Resources:** Oscar Vaes, Claire Detrain.

**Software:** Oscar Vaes.

**Supervision:** Claire Detrain.

**Validation:** Oscar Vaes, Claire Detrain.

**Visualization:** Oscar Vaes.

**Writing – original draft:** Oscar Vaes, Claire Detrain.

**Writing – review & editing:** Oscar Vaes, Claire Detrain.

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
