## [Decision Letter · Decision Letter 0]

12 May 2022

PONE-D-22-09396Colony specificity and starvation-driven changes of activity patterns in the red ant Myrmica rubraPLOS ONE

Dear Dr. Vaes,

Thank you for submitting your manuscript to PLOS ONE. After careful consideration, we feel that it has merit but does not fully meet PLOS ONE’s publication criteria as it currently stands. Therefore, we invite you to submit a revised version of the manuscript that addresses the points raised during the review process.

We look forward to receiving your revised manuscript.

Kind regards,

Alessandro Cini

Academic Editor

PLOS ONE

Journal Requirements:

Additional Editor Comments:

Dear authors,

your ms has been review by two reviewers and by myself.

We overall agree that the topic is interesting, the experimental design as well as the data-analyses are overall correct and that the paper is generally well-written.

We are thus very positive about the paper, however I invite you to carefully address the points raised by the two reviewers (see below).

While many of them are minor corrections, a couple of points needs a proper explanation and/or amendement.

Namely:

1) the point raised by rev1 about the metric used to estimate synchronization. I agree with it: you should better justify it and convince that CV is a trustable metric. I am not so convinced, and this is a rather significant issue, if not solved.

2) (my comment) . You measured activity by quantifying the number of pixels that changed from black to white (and viceversa). And you say that this is not influenced by the number of ants present. If so, when you observe the lack of difference in acitivty levels (for example in the foraging area during starvation) you could not exclude that, apart form activity, there was a different number of workers, am I right?

I.e. as far as I understand is not possible form your data tom understand if starvation led to a recruitment of workers to the foraging area...isn't it? If I am right, please add a discussion about that, if I am not, please explain it better (to me and in the ms!)

Waiting fro your revision,

best regards,

Alessandro Cini

Reviewers' comments:

Reviewer's Responses to Questions

**Comments to the Author**

1. Is the manuscript technically sound, and do the data support the conclusions?

Reviewer #1: Yes

Reviewer #2: Yes

2. Has the statistical analysis been performed appropriately and rigorously? 

Reviewer #1: Yes

Reviewer #2: Yes

3. Have the authors made all data underlying the findings in their manuscript fully available?

Reviewer #1: Yes

Reviewer #2: Yes

4. Is the manuscript presented in an intelligible fashion and written in standard English?

Reviewer #1: Yes

Reviewer #2: Yes

5. Review Comments to the Author

Reviewer #1: This reports on very interesting work.

Readers will appreciate the clear explanation of the methods. I found the Discussion section more difficult to understand and this would benefit from editing for English grammar and style.

I don't see why a high CV necessarily indicates synchronization. I am interpreting 'synchronization' to mean that at a given time the ants show some similarity in the extent of activity as in lines 239-40. I can see that if in some observations all ants were more active, and in others some ants were less active, this would increase CV. But isn't it possible that CV could increase without synchronization? That is if the distribution of activity varies among observations it could lead to different values across the day, even though in one particular observation all the ants are not showing the same activity level?

The manuscript is written as if the authors expected all colonies to be similar, but the main result, that colonies differ in activity level, is interesting. The discussion should consider how these differences among colonies influence the ecology of Myrmica rubra. Colony variation in activity has been found in other species, and the literature on this should be reviewed. For example, harvester ant colonies differ in foraging and patrolling activity (Gordon et al 2011 Behav Ecol), and this is associated with differences among colonies in gene expression (Friedman et al 2020 Commun Biol). Fire ant colonies differ in foraging activity, and this varies across regions (Bockhoven al PLOS One 2015). There are many other examples for ants, and colony variation in activity has been seen in many hymenopteran species, e.g. in wasps (Monceau et al Ins Soc 2015); in honey bees it is well known (e.g. Wray et al Anim Behav 2011) and can be artificially selected (Hunt et al 1995 Genetics). The interpretation of the results does not seem to take into account the possibility that activity influences further activity, because activity determines encounter rate, which then influences further movement. For example, if activity is synchronized, it seems likely to be due to interactions, not just to spontaneous increases or decreases in activity that all the ants experience at the same time.

I did not understand the idea on lines 574-78 and elsewhere that cycles of high activity constrain information flow but high activity increases efficiency. I think information flow comes from interactions? so is the reason cycles decrease information flow that oscillations would include periods of reduced interaction? - but wouldn't they also include periods of high interaction? And how does high activity increase efficiency if not through interactions?

The changes in activity in response to starvation are consistent with the association of movement and encounters. Increased encounter rates might facilitate recruitment to a new food source if it appears, or, for workers that are feeding the brood, increased movement would improve the chances of finding other ants that have food.

Reviewer #2: This study investigated colony-level activity in Myrmica rubra, focusing on the effect of food deprivation. The author recorded colony-level activity for 15 days, both outside and inside the nest. They found that activity levels varied across colonies and reacted to starvation. Also, they analyze the pattern of bursts and daily activities. The treatment of nutrition availability is novel and important for understanding the temporal organization of ant colonies. Also, the authors analyzed time-series data very well. I would recommend that this paper is accepted for publication with minor revisions.

Line 45-56: I didn’t understand the connection between the “lazy” ant and the current purpose of this study well. The authors investigated 1) colony-level activity, 2) synchronization (burst/coefficient), and 3) periodicity but not the inactivity. I would suggest revising the introduction or adding analysis for inactivity (*individual-level analysis is needed). The discussion part makes sense to me.

Line 71-73: I would recommend adding about the effect of starvation/nutrient state on ant behaviour around here. What happens when ant colonies are starved, or an individual is hungry? There are many previous studies investigating the effect of starvation on ant colonies. I think the treatment of starvation is the core of this study, but I feel not enough to review previous studies.

Line 97: Can you explain more about the big difference between new(?) species and previous species? Why or how these traits are important in characterizing the stability and the colony-specificity of the activity patterns? It is better to mention Line 500-502 in the introduction.

I wonder whether the activity level was consistent inside and outside the nest in the same colony. For example, if activity inside the colony is high, is their activity outside also high?

Figure 3: I would suggest changing figure 3 as figure 3(a) and figure 4 to figure 3 (b) to reduce redundancy.

Figures 3,4,6,7: I think this is not evolution. You can say time evolution.

Figure 5: Is 5c colony 7?

Figure 8: I am not sure why the authors provided fig 8. Figure 8 can be supp. fig. or give all that periodicity was detected.

Figure 10: be careful; this image is very low resolution.

Supp. The tables were all images (looks like a screenshot). The authors should change them to tables and change column names.

6. PLOS authors have the option to publish the peer review history of their article (what does this mean?). If published, this will include your full peer review and any attached files.

Reviewer #1: No

Reviewer #2: No

---

## [Author Response · Author response to Decision Letter 0]

10 Jun 2022

Dear authors,

your ms has been review by two reviewers and by myself.

We overall agree that the topic is interesting, the experimental design as well as the data-analyses are overall correct and that the paper is generally well-written.

We are thus very positive about the paper, however I invite you to carefully address the points raised by the two reviewers (see below).

While many of them are minor corrections, a couple of points needs a proper explanation and/or amendment.

Namely:

2) (my comment). You measured activity by quantifying the number of pixels that changed from black to white (and viceversa). And you say that this is not influenced by the number of ants present. If so, when you observe the lack of difference in acitivty levels (for example in the foraging area during starvation) you could not exclude that, apart form activity, there was a different number of workers, am I right?

I.e. as far as I understand is not possible form your data tom understand if starvation led to a recruitment of workers to the foraging area...isn't it? If I am right, please add a discussion about that, if I am not, please explain it better (to me and in the ms!)

Waiting fro your revision,

best regards,

Alessandro Cini

Authors response: We would like to thank the editor as well as the two reviewers for their constructive comments and insightful suggestions. We have paid special attention to address all the questions that were raised and to improve the manuscript wherever needed.

Point-by-point answers are provided below

Editor’s comment 

1) the point raised by rev1 about the metric used to estimate synchronization. I agree with it: you should better justify it and convince that CV is a trustable metric. I am not so convinced, and this is a rather significant issue, if not solved.

Authors response: This first comment about the CV metrics is in line with reviewer 1’s question. Please look at our detailed answers to referee 1.

2) (my comment). You measured activity by quantifying the number of pixels that changed from black to white (and viceversa). And you say that this is not influenced by the number of ants present. If so, when you observe the lack of difference in acitivty levels (for example in the foraging area during starvation) you could not exclude that, apart form activity, there was a different number of workers, am I right?

I.e. as far as I understand is not possible form your data tom understand if starvation led to a recruitment of workers to the foraging area...isn't it? If I am right, please add a discussion about that, if I am not, please explain it better (to me and in the ms!)

Authors response: The editor is right when saying that we cannot directly deduce the number of ants from the number of changing pixels. Indeed, as stated at lines 184 to 190 for the activity inside the nest, during the quite stable condition of colony satiation, we found no correlation between the number of ants that were counted inside the nest at 9am and the number of moving pixels detected at this time point (See S1). Likewise, for the ants foraging in the outside, we also counted their number at 9am for each experimental day of the satiation period. Further to the editor’s comment, we looked for and found no correlation between these numbers of foragers observed on each day and the extra-nest activity, i.e., the number of moving pixels, that was measured at the same time point (9 am). This information was added at line 188 and in S1. 

As for the impact of starvation on the recruitment of foragers, one cannot draw some trend from the activity data alone that are provided in the MS. So, we looked into the counts of the number of foragers that were made on each morning at 9 am. We ran an ANOVA to check whether the number of foragers was influenced by the colony, the phase of the experiment (Satiation VS Starvation VS recovery) and/or the interaction between the two factors. Results showed a significant colonial effect on the number of ants on the foraging area but did not show any effect of the phase nor interaction effect. This means that we found no significant effect of starvation on the recruitment of workers to the foraging area.

Reviewers' comments:

Reviewer #1: This reports on very interesting work.

Readers will appreciate the clear explanation of the methods. I found the Discussion section more difficult to understand and this would benefit from editing for English grammar and style.

Authors response: Thank you for this positive comment. Following your suggestion, we asked to a professional English-editing service to correct the discussion as well as other parts of the revised MS. We have the feeling that these changes have clarified the discussion and have improved the grammar and style of the MS. Please note that the title was slightly changed.

R #1: I don't see why a high CV necessarily indicates synchronization. I am interpreting 'synchronization' to mean that at a given time the ants show some similarity in the extent of activity as in lines 239-40. I can see that if in some observations all ants were more active, and in others some ants were less active, this would increase CV. But isn't it possible that CV could increase without synchronization? That is if the distribution of activity varies among observations it could lead to different values across the day, even though in one particular observation all the ants are not showing the same activity level?

Authors response: Synchronization can be challenging to quantify, especially when it is not coupled with rhythmicity as it may be the case for some species showing short-term activity cycles (STACs). However, synchronization also occurs when many ants are active/inactive simultaneously, even though these activity/inactivity bouts do not take place at regular time intervals. Thus, we looked for metrics that best expressed the amplitude of fluctuations around the average activity level. In literature, the index of dispersion (ratio of variance over mean) proved to be a trustful way to estimate synchronization of activity in locusts (Despland and Simons 2006) and ants (Cole 1992, Doering et al. 2019). It is even used in neuroscience where it is referred to as the Fano factor (Stevens and Zador 1998). Being directly derived from this index of synchronization, we used the coefficient of variation (CV) in our paper. When nestmates are synchronized, this generates bouts of activity/inactivity that were highly fluctuating (negatively or positively) around the mean activity value. This resulted in an increase of the standard deviation of activity values that were measured on this day and hence the associated CV value. 

That being said, we fully agree with reviewer 1 that a high value of CV might also reflect many fluctuations of small amplitude taking place all over the course of the experimental day. In this latter case, fluctuations are more pertaining to some “activity noise” rather than to a marked synchronization of workers’ activity. That is the reason why we did not limit our analysis of synchronization to the CV metric but we complemented it by a peak analysis. This peak analysis took into account threshold values in activity changes across successive observations. This approach enabled us to discriminate actual bouts of activity (and therefore actual events of synchronization) from frequent but small and “noisy” variations of activity. Furthermore, this peak analysis showed that the number of detected activity peaks did not change across the experimental phases. This indicates that the increased CV values measured during starvation were not due to a higher rate of peak occurrence but resulted from activity bouts of higher amplitude that were generated by a higher synchronization among nestmates.

Moreover, we added a citation (line 239) from Cole’s paper (1992) as an extra reference for a successful measure of synchronization of ants’ activity with this index. We addressed the interesting point raised by the reviewer on alternative explanations for high CV values on lines 243 to 247.

R #1: The manuscript is written as if the authors expected all colonies to be similar, but the main result, that colonies differ in activity level, is interesting. The discussion should consider how these differences among colonies influence the ecology of Myrmica rubra. Colony variation in activity has been found in other species, and the literature on this should be reviewed. For example, harvester ant colonies differ in foraging and patrolling activity (Gordon et al 2011 Behav Ecol), and this is associated with differences among colonies in gene expression (Friedman et al 2020 Commun Biol). Fire ant colonies differ in foraging activity, and this varies across regions (Bockhoven al PLOS One 2015). There are many other examples for ants, and colony variation in activity has been seen in many hymenopteran species, e.g. in wasps (Monceau et al Ins Soc 2015); in honey bees it is well known (e.g. Wray et al Anim Behav 2011) and can be artificially selected (Hunt et al 1995 Genetics). 

Authors response: Colony variation in activity was briefly evoked in the first submitted MS by namely citing the work of Gordon (2013) and (Bockoven et al. 2017). However, we agree with the reviewer that examples of colony specificity could be enlarged to other Hymenoptera species and should be better highlighted in the revised MS. Therefore, we added the references that were kindly suggested by referee 1 in the introduction (see line 69-71 in the revised MS). In the discussion, we relate our results to previous reports about colony variation in foraging activity in other ants (see lines 485-488). Finally, we emphasize on differences in gene expression and in genetic diversity that are associated with differences in foraging activity (lines 493-497). The impact of activity variation on colony fitness is further supported by the references suggested by referee 1 at line 603-610. These lines are showing how the variable the fitness consequences may be, depending on the studied species and its ecology. 

R #1: The interpretation of the results does not seem to take into account the possibility that activity influences further activity, because activity determines encounter rate, which then influences further movement. For example, if activity is synchronized, it seems likely to be due to interactions, not just to spontaneous increases or decreases in activity that all the ants experience at the same time.

Authors response: We fully agree that activity can further enhance activity, through increasing encounter rates and mobility. Mutual activation between ants, that has been extensively studied by B.J. Cole, is a widely accepted explanation as to why activity can be synchronized to some degree between workers. This is the reason why the discussion does not go into details about the mechanisms behind activity variations at the group level. Synchronization of activity as being a self-amplifying phenomenon is nevertheless stated in lines 78 to 81 of the introduction and lines 548 to 550 in the discussion. If required, we may develop this idea in more details, but we hope these lines meet referee 1’s concern. 

R #1: I did not understand the idea on lines 574-578 and elsewhere that cycles of high activity constrain information flow but high activity increases efficiency. I think information flow comes from interactions? so is the reason cycles decrease information flow that oscillations would include periods of reduced interaction? - but wouldn't they also include periods of high interaction? And how does high activity increase efficiency if not through interactions?

Authors response: Thank you for bringing our attention to this part of the discussion. We definitely agree with referee 1 that information usually comes from interactions. The contacts themselves can convey information with or without a transfer of chemical cues (e.g., colony odor). Contact-based information can be transmitted by ritualized tactile motor displays or incidental kinetic encounters. However, information can also be embedded in the environment such as the chemical trails laid on the substrate by successful foragers. 

To account for the propagation of information flow, one should include not only the process of information transmission (enhanced when activity/encounters are peaking) but also the process of information loss (e.g., through an informed ant forgetting relevant information, or, through the decay of a chemical information). The interplay between these two mutually antagonistic feedback loops may lead to the population of informed ants either declining to zero or growing until a steady state. The idea that rhythmicity of activity may constrain information flow outcomes from a data-driven model developed by Richardson et al. (2017). He studied short-term activity cycles (STACs) in Temnothorax ant species and built a model, derived from epidemiology SIS models, with different information transmission - and loss - rate combinations. Their simulations revealed that regularly-oscillating bouts of activity – and hence oscillating bout of interactions, could impede information flow. This typically occurs when the lifetime of the information is shorter than the duration of the quiescent periods of activity, then information loss during quiescent periods dominates over information spread during active periods. We found Richardson’s predictions worthwhile to be discussed in our paper, even though they still need to be experimentally supported by comparative studies. 

To meet reviewer 1 comment, without entering into too many details, we better explain how the interplay of transmission/loss of information and oscillations with periods of reduced interactions may influence the propagation of information. This was done at lines 558-567.

R #1: The changes in activity in response to starvation are consistent with the association of movement and encounters. Increased encounter rates might facilitate recruitment to a new food source if it appears, or, for workers that are feeding the brood, increased movement would improve the chances of finding other ants that have food.

Authors response: We agree that increased encounter rates might help to generate a faster response from the group. However, starvation is a challenging state for the colony where energy needs to be spent in an efficient way. The concurrent need for energy saving and for increased interactions to facilitate recruitment to a new food source are both matched by the low overall activity coupled to high bursts of activity that we observed in our experiment. Following reviewer 2’s comment, we added in lines 63 to 68 of the introduction, complementary information about the effect of starvation on the activity observed inside the nest as well as on the foraging area. 

Reviewer #2: This study investigated colony-level activity in Myrmica rubra, focusing on the effect of food deprivation. The author recorded colony-level activity for 15 days, both outside and inside the nest. They found that activity levels varied across colonies and reacted to starvation. Also, they analyze the pattern of bursts and daily activities. The treatment of nutrition availability is novel and important for understanding the temporal organization of ant colonies. Also, the authors analyzed time-series data very well. I would recommend that this paper is accepted for publication with minor revisions.

Authors response: We would like to thank reviewer 2 for these positive comments.

R #2: Line 45-56: I didn’t understand the connection between the “lazy” ant and the current purpose of this study well. The authors investigated 1) colony-level activity, 2) synchronization (burst/coefficient), and 3) periodicity but not the inactivity. I would suggest revising the introduction or adding analysis for inactivity (*individual-level analysis is needed). The discussion part makes sense to me.

Authors response: We aimed to draw a general picture of current knowledge about “activity” and its “inactivity” counterpart, both at the individual and collective level. However, we acknowledge that too much emphasis was put on “lazy” ants that were not the purpose of this study. Therefore, this paragraph (lines 45 to 47) has been significantly reduced (117 words to 34 words) in the revised MS.

R #2: Line 71-73: I would recommend adding about the effect of starvation/nutrient state on ant behaviour around here. What happens when ant colonies are starved, or an individual is hungry? There are many previous studies investigating the effect of starvation on ant colonies. I think the treatment of starvation is the core of this study, but I feel not enough to review previous studies.

Authors response: Many thanks for this suggestion. We agree that the effect of starvation on ants’ activity is central in our study. There is actually a huge amount of data about the impact of starvation on ant colonies. Hunger can influence ant colonies in many different ways. For instance, one can cite an impact on food selectivity and feeding duration (Josens and Roces 2000), on the spatial organization of individuals inside the nest (Mailleux et al. Ins Soc 2011), the division of tasks (Blanchard et al. 2000), on the distribution of food within the colony (Howard and Tschinkel 1980). Starvation can also have an impact on the propensity to care for brood (Cassill and Tschinkel 1999), on the foraging effort (Traniello 1977, Franks et al. 1990, Mailleux et al. Ethology 2010) as well as on the recruitment decision rules (Mailleux et al. J. Exp. Biol. 2006). In order to address referee 2 comment, we have restricted the additional information given in the revised MS to these last impact of starvation that have direct consequences on the activity inside and outside the nest. Indeed, we feel that the paper is already quite long but we are ready to further develop this topics if required by the reviewer or the editor. This was done in the introduction in lines 63 to 68. 

R #2: Line 97: Can you explain more about the big difference between new(?) species and previous species? Why or how these traits are important in characterizing the stability and the colony-specificity of the activity patterns? It is better to mention Line 500-502 in the introduction.

Authors response: We were referring to M. rubra as a “new” ant species by comparison to the species cited in the former paragraph, in lines 93 to 96. What is “new” is not the species itself but the fact that our paper is the first one to characterize M. rubra activity patterns. To avoid any confusion, we removed the word “new” from line 94.

As for the main differences between M. rubra and the previously studied ants from Leptothorax and Temnothorax genera, one can cite their tempo and colony size. The latter ones are characterized by small sized colonies, slow paced movements and they often display short-term activity cycles (STACs) with periodic activity bursts every 10 to 30 minutes. Many hypotheses were made about the adaptive value of such activity fluctuations but comparative studies on other ant species were lacking to support some generic value of these activity cycles. By studying M. rubra species, we found that such activity cycles are not observed in an ant colony with a higher tempo and hosting a much larger population of workers. 

As suggested by the reviewer, the variability in workers’ genetic relatedness as a M. rubra trait has been added in the introduction at lines 95 and 96. Because it is directly related to the argument developed, we have also evoked this variation in genetic relatedness in lines 497 to 499 from the discussion. 

We hope these changes will contribute to clarify our approach.

R #2: I wonder whether the activity level was consistent inside and outside the nest in the same colony. For example, if activity inside the colony is high, is their activity outside also high?

Authors response: Thanks for this interesting question. We analyzed the relationship between the activity inside and outside the nest for each colony separately. The results are now reported in a revised paragraph in lines 348 to 351 as well as in S2. In most cases, we found that high the activity indices within the nest of a certain colony did not correlate with high (or low) activity indices in the foraging area. 

R #2: Figure 3: I would suggest changing figure 3 as figure 3(a) and figure 4 to figure 3 (b) to reduce redundancy.

Authors response: This has now been done and all figure numbers were adapted in the manuscript to match this change. Thanks for the suggestion. 

R #2: Figures 3,4,6,7: I think this is not evolution. You can say time evolution.

Authors response: You are right. Changes were made on lines 318, 400 and 424.

R #2: Figure 5: Is 5c colony 7?

Authors response: Fig. 5C represents the same data as 5B but with “smoothed” values. Since the legend of the figure was ambiguous, changes were made on line 369 to make it clearer. Thank you for pointing this out.

R #2: Figure 8: I am not sure why the authors provided fig 8. Figure 8 can be supp. fig. or give all that periodicity was detected.

Authors response: Fig 8 is displayed in the MS to help the reader visualize how wavelet analysis works and how periodicity can be detected. This figure also provides a direct comparison of two typical activity patterns shown by the same colony but corresponding to different phases of the experiment. We think that showing these two examples is a good compromise for the reader. It helps to understand and visualize the concepts of periodicity while not distracting from the main results. For this reason, we would like to keep them in the MS as we feel it facilitates the understanding of the MS.

R #2: Figure 10: be careful; this image is very low resolution.

Authors response: The resolution of the image was increased.

R #2: Supp. The tables were all images (looks like a screenshot). The authors should change them to tables and change column names.

Authors response: The format of these supplementary materials was changed as well as their column names and general readability. We took the opportunity to group all Tukey HSD post hoc tests into one table and to group pairwise t-tests that were performed on different phases but for the same metric. For example, all the t-tests on the daily activity inside the nest (S3 on phase 2 and S4 on phase 3) are now grouped under a single table. The legends associated to each supplementary material was adapted accordingly. Thank you for this suggestion.

---

## [Decision Letter · Decision Letter 1]

3 Aug 2022

Colony specificity and starvation-driven changes in activity patterns of the red ant Myrmica rubra

PONE-D-22-09396R1

Dear Dr. Vaes,

We’re pleased to inform you that your manuscript has been judged scientifically suitable for publication and will be formally accepted for publication once it meets all outstanding technical requirements.

Kind regards,

Alessandro Cini

Academic Editor

PLOS ONE

Additional Editor Comments (optional):

Dear authors,

you made a good effort in addressing all the raised issues and I believe the ms can now be accepted for publication, please just note the below comment from one of the reviewers.

my best regards

Alessandro Cini

Line 185 The number of foragers is unclear. That means the number of ants in foraging area, outside the nest, or ants that were foraging (eating, milling and/or currying food)? I think that foraging area and outside the nest are the same meaning in your setup (Figure 1).

Reviewers' comments:

Reviewer's Responses to Questions

**Comments to the Author**

1. If the authors have adequately addressed your comments raised in a previous round of review and you feel that this manuscript is now acceptable for publication, you may indicate that here to bypass the “Comments to the Author” section, enter your conflict of interest statement in the “Confidential to Editor” section, and submit your "Accept" recommendation.

Reviewer #1: All comments have been addressed

Reviewer #2: All comments have been addressed

2. Is the manuscript technically sound, and do the data support the conclusions?

Reviewer #1: Yes

Reviewer #2: Yes

3. Has the statistical analysis been performed appropriately and rigorously? 

Reviewer #1: Yes

Reviewer #2: Yes

4. Have the authors made all data underlying the findings in their manuscript fully available?

Reviewer #1: Yes

Reviewer #2: (No Response)

5. Is the manuscript presented in an intelligible fashion and written in standard English?

Reviewer #1: Yes

Reviewer #2: Yes

6. Review Comments to the Author

Reviewer #1: (No Response)

Reviewer #2: The authors did a great job. The manuscript has been revised according to my suggestion (rev. 2). The answers to all comments of the editor and reviewer 1 are also clear for me.

The current manuscript is accepted for publication.

Just one suggestion:

Line 185 The number of foragers is unclear. That means the number of ants in foraging area, outside the nest, or ants that were foraging (eating, milling and/or currying food)? I think that foraging area and outside the nest are the same meaning in your setup (Figure 1).

7. PLOS authors have the option to publish the peer review history of their article (what does this mean?). If published, this will include your full peer review and any attached files.

Reviewer #1: No

Reviewer #2: No

---

## [Editor Report · Acceptance letter]

4 Aug 2022

PONE-D-22-09396R1 

Colony specificity and starvation-driven changes in activity patterns of the red ant *Myrmica rubra*

Dear Dr. Vaes:

I'm pleased to inform you that your manuscript has been deemed suitable for publication in PLOS ONE. Congratulations! Your manuscript is now with our production department. 

Kind regards, 

on behalf of

Dr. Alessandro Cini 

Academic Editor

PLOS ONE